# Increased cancer risk in HIV-infected individuals occupationally exposed to chemicals: Depression of p53 as the key driver

Donald C. Udah[1,6], Adeleye S. Bakarey[2,7], Gloria O. Anetor[3], Maxwell Omabe[4], Victory F. Edem[5], Olusegun G. Ademowo[2,8], John I. Anetor[1] *

1 Department of Chemical Pathology, Laboratory for Toxicology and Micronutrient Metabolism, College of Medicine, University of Ibadan, Ibadan, Nigeria, 2 Institute for Advanced Medical Research and Training, College of Medicine, University of Ibadan, Ibadan, Nigeria, 3 Department of Public Health Science, Faculty of Health Sciences, National Open University of Nigeria (NOUN), Abuja, Nigeria, 4 Department of Medical Laboratory Sciences, School of Biomedical Science, Faculty of Health Science, Ebonyi State University, Nigeria, 5 Department of Immunology, University of Ibadan, Ibadan, Nigeria, 6 JSI Research & Training Institute Inc. (JSI), Abuja, Nigeria, 7 Department of Biomedical Laboratory Science, College of Medicine, University of Ibadan, Ibadan, Nigeria, 8 Department of Pharmacology and Therapeutics, College of Medicine, University of Ibadan, Ibadan, Nigeria

* johnanetor@gmail.com

**Data Availability Statement:** Our research data is available to all interested parties on the Open Science Framework (OSF) repository, which allows

## Abstract

The growing exposure to occupational chemicals and the spread of human immunodeficiency virus (HIV) infection are major global health issues. However, there is little data on the carcinogenic risk profile of HIV-infected individuals who have been occupationally exposed to chemical mixtures. This study therefore investigated the levels of cancer risk biomarkers in HIV-infected individuals exposed to occupational chemicals, exploring the relationship between apoptotic regulatory and oxidative response markers as a measure of cancer risk. Study participants (mean age 38.35±0.72 years) were divided into four groups according to their HIV status and occupational chemical exposure: 62 HIV-positive exposed (HPE), 66 HIV-positive unexposed (HPU), 60 HIV-negative exposed (HNE), and 60 HIV-negative unexposed (HNU). Serum p53, β-cell lymphoma-2 (bcl2), 8-hydroxydeoxyguanosine (8-OHdG), superoxide dismutase (SOD), and malondialdehyde (MDA) were measured using standard methods. Clusters of differentiation 4 (CD4$^+$) T-lymphocytes were enumerated using flow cytometry. Serum p53 and bcl2 levels in HPE (0.91±0.11ng/ml and 122.37±15.77ng/ml) were significantly lower than HNU (1.49±0.15ng/ml and 225.52±33.67ng/ml) (p < 0.05), respectively. Wildtype p53 and bcl2 were positively and significantly correlated with 8-OHdG (r = 0.35, p<0.001; r = 0.36, p<0.001) and SOD (r = 0.38, p<0.001; r = 0.39, p<0.001). After controlling for gender, age, BMI, and cigarette smoking, both HIV status and SOD activity were significantly associated with wildtype p53 and bcl2 (p < 0.05). Malondialdehyde was significantly higher in the HPE (0.72 ± 0.01 mg/ml) than in the HNE (0.68 ± 0.01 mg/ml) and HNU (0.67 ± 0.01 mg/ml) groups (p < 0.05). The HPE group showed significantly lower CD4 counts than the HNE and HNU groups. Individuals who are HIV-infected and occupationally exposed to chemicals have a constellation of depressed immunity, elevated oxidative stress, and loss of tumour suppressive functions, which together intensify cancer

for unrestricted data use. The link to access the dataset is provided below: https://osf.io/vj6wp/?view_only=ff2c2c4cf7024a42a0452531ad6902c1.

**Funding:** The authors received no specific funding for this work.

**Competing interests:** The authors have declared that no competing interests exist.

risk, providing valuable scientific and public health bases for preventive measures in this vulnerable population.

## Introduction

Epidemiological studies have demonstrated that patients with compromised immune systems, such as those seen with HIV infections, have a higher cancer incidence rate [1–3]. In addition, patients with Acquired Immunodeficiency Syndrome (AIDS) defining cancers, such as Kaposi sarcoma and non-Hodgkin's lymphoma, had a poor prognosis prior to the introduction of Highly Active Antiretroviral Therapy (HAART), with a median survival of 5 to 6 months [4]. The increased availability of HAART has largely contributed to the reduction in the risk of AIDS-defining cancers, allowing HIV-infected individuals to live longer [5]. This has resulted in reduced incidence of malignancies which are linked to AIDS and weakened immunity [6]. However, the occurrence of non-AIDS-defining malignancies is increasing among people infected with HIV [7].

Although the mechanisms underlying the increased risk of cancer among HIV-infected individuals remain poorly understood, they may be associated with non-HIV factors such as occupational exposure to toxic metals. Thus, HIV infection and occupational exposure to hazardous chemicals are two risk factors that may be associated with the disruption of immunological and/or non-immune surveillance systems against cancer, especially the p53-dependent metabolic cascade. The tumor suppressor p53 and the anti-apoptotic protein bcl2 were two of the first cancer genes identified, and their interaction is critical for cancer biology [8].

Many people, including those living with HIV, work in a variety of occupations. Some of these jobs are well-known to expose workers to toxic metals like lead, mercury, chromium, and cadmium; volatile organic compounds (VOCs) including benzene, particularly from sources such as motor-vehicle exhaust and gasoline vapor emissions; and skin-irritating substances such as oils, greases, solvents, and detergent [9–12]. Exposure to chemical mixtures, including toxic metals, is a common risk factor for cancer, especially among certain occupational groups. This is primarily because various substances prevalent in industrial and agricultural environments have been identified as carcinogenic [13–17]. These chemicals can cause DNA damage, resulting in mutations and abnormal cellular behaviour. Additionally, genetic and epigenetic alterations are known to predispose a cell clone to malignancy [18]. Multiple defence mechanisms, both immunological and non-immune surveillance system, may be employed to ward off this threat [19]. The best-known form of genetic surveillance against cancer acts through the p53-dependent pathway, with the p53 tumour suppressor gene dubbed the "Guardian of the Genome" [20]. It prevents genomic instability and suppresses tumourigenesis through regulation of diverse cellular processes including cell proliferation, DNA repair, and cell death [21].

The wildtype p53 protein binds to DNA, specifically the promoter region of p21, which is a cyclin-dependent kinase inhibitor, halting the cell cycle in the G1 phase [22]. This cell cycle arrest allows time for cells to repair DNA damage caused by cytotoxic stressors [23]. According to Leroy et al., [24], more than 50% of all human tumours carry p53 mutations. These mutations impair the DNA-binding capacity of the p53 protein. Consequently, cells with damaged DNA but with maintained mitotic capacity go on dividing and increasing the risk of cancer. Furthermore, the anti-apoptotic protein bcl2 modulates programmed cell death. It inhibits apoptosis, thereby promoting cell survival. Dysregulation of bcl2 expression has been

implicated in carcinogenesis. Since p53 and bcl2 play important roles in controlling the cell cycle and apoptosis, they may be useful biomarkers for cancer risk surveillance. However, there are paucity of studies that evaluate the p53 and bcl2 proteins in HIV patients who are occupationally exposed to chemical mixtures. The present study aimed to evaluate whether levels of apoptotic regulatory markers p53 and bcl2 are associated with HIV infection, occupational exposure, oxidative damage, and antioxidant protection.

## Materials and methods

### Study population

This was a cross-sectional study conducted in Nigeria involving 128 HIV positive and 120 HIV negative consenting adults, with a mean age of 38.35 ± 0.72 years. Apparently healthy HIV-positive and HIV-negative adults were recruited from both the community and a health facility between April 2017 and October 2017. The study was conducted at Dr. Lawrence Henshaw Memorial Hospital Calabar, with community outreach in Calabar Municipal and Calabar South Local Government Area. Individuals who tested HIV negative in the community and those who visited the HIV Testing and Counselling units at the health facility for a voluntary HIV testing service and tested negative were matched for age, sex, and similarity in environmental as well as occupational characteristics with HIV positive cases, and all recruited into the study.

A structured questionnaire was administered to obtain relevant information, including occupational exposure data. Prior to its use in the study, the questionnaire was pilot tested with a small group representative of the study population to ensure clarity and relevance. Additionally, a panel of experts in toxicology and environmental epidemiology reviewed the questionnaire to ensure its validity. Based on HIV status and occupational chemical exposure, the study population (N = 248) was divided into 4 groups: HIV-positive exposed (HPE; n = 62), HIV-positive unexposed (HPU; n = 66), HIV negative exposed (HNE; n = 60), and HIV negative unexposed (HNU; n = 60). The study specifically examined two distinct groups: individuals who were occupationally exposed to chemicals and an unexposed category. The occupationally exposed cohort, consistent with previous studies, had worked in professions associated with toxic metals and hazardous chemicals. These occupations included Cement factory worker [25, 26], Electrician [27, 28], Bars & Night clubs worker [29, 30], Mechanic [9, 31], Painter & Printing press worker [9, 10], Weed sprayer & Pesticides worker [11, 32, 33], Welder [12], and Petrol Station Dispenser [34]. Therefore, the occupationally exposed HIV-infected and uninfected participants were exposed to toxicants including automotive chemicals, cement dust, diesel exhaust, electronic waste, fungicides and pesticides, metal cutting fluids, metal fumes, second hand smoke, solvents, sulfuric acid, and wood dust. Furthermore, the unexposed category includes those who work in vocations that are considered low risk for toxic metal exposure, such as teachers, students, and administrative personnel in both the commercial and public sectors (data entry clerks, customer service clerks, receptionists).

Participants with a history of hereditary immune deficiency, primary immune deficiency, autoimmune disease, immunosuppressive drug use, previous cancer diagnosis, supplement use at the time of the study or within the month preceding it, refusal to consent, or age <18 years were excluded from this study. Participants were consecutively identified and recruited into the study. The HIV status of participants in this study was confirmed using the national HIV testing algorithm [35].

## Ethics statement

Ethical approval for this study was received from the Ethics Committee of the University of Ibadan/University College Hospital (UI/EC/16/0226). Additionally, written informed consent was obtained from all the participants selected.

## Anthropometric measurements

Body weight (Kg) was measured using an electronic scale with an accuracy of 0.1 Kg, and height was measured using a stadiometer with an accuracy of 0.1 Cm. Prior to the study, we used standard weights and known height measurements to calibrate the weighing scale and stadiometers. We performed regular calibration checks throughout the study to maintain measurement accuracy and detect any drifts in the scale's performance. We assessed inter-rater and intra-rater reliability by having multiple research assistants measure height and body weight and comparing the results. Participants were instructed to take off their shoes, coats, and hats (if they were wearing them). The participants' heights were measured while they stood straight against the stadiometer, with their heels, buttocks, shoulders, and heads touching the vertical surface. During weight measurement, participants were instructed to remain still in the centre of the weighing scale. Each participant's body mass index (BMI) was calculated as weight in kilograms divided by the square of the height in meters ($kg/m^2$).

## Sample collection

Blood samples were aseptically obtained from each participant using disposable, pyrogen-free, vacutainer needles. Whole blood (5ml) was collected into serum separator tubes (SST) and 2ml into k3-EDTA vacutainers. Samples collected into SST were allowed to clot in an upright position for at least 30 minutes but not longer than 1 hour before centrifugation at 2500 revolutions per minute (rpm) for 15 minutes. The serum was aliquoted into cryogenic vials while other samples not immediately required were stored at -86˚C until ready for analysis.

## Quantification of CD4+ lymphocytes

The Becton Dickinson (BD) FASCount system (Becton, Dickinson and Company, California, USA) was used to identify and count CD4+ T cells. This flow cytometry for CD4 enumeration employs specific monoclonal antibodies tagged with fluorescent dyes to bind CD4+ cells [36]. These cells are then passed through a fluorescence-activated cell sorter (FACS), where they are identified and counted based on their fluorescence intensity. A single test requires one ready-to-use reagent tube containing specific monoclonal antibodies labelled with fluorochromes. After adding whole blood (5 μL) to the reagent tube, fluorochrome-labelled antibodies in the reagents bind specifically to CD4 lymphocyte surface antigens. The instrument runs the sample after adding a fixative solution to the reagent tubes, passing individual cells through a laser beam that excites them to emit light signals. This fluorescent light provides the information required for cell quantification and sorting. The reagent tubes also contain a known number of fluorochrome-integrated reference beads, which serve as a fluorescence standard for identifying and quantifying cells by the fluorescence-activated cell sorter. Samples were analyzed within 6 hours of collection. The Centers for Disease Control and Prevention (CDC) classification system of HIV-infected individuals were used to further categorize the study population into three groups: CD4 counts (a) $\geq$ 500 cells/μL; (b) 200–499 cells/μL and (c) <200 cells/μL [37].

## Determination p53, bcl2, SOD and 8OHdG

The serum levels of p53, bcl2, SOD and 8OHdG were determined using the ELISA method in accordance with the manufacturer's procedure (Melsin Medical Co., Limited, China). These kits are based on the principle of a one-step double-antibody sandwich enzyme-linked immunosorbent assay [38]. First, a specific antibody that is immobilized on a plate binds to the target antigen. Then, a second enzyme-linked antibody is added and binds to it, creating a sandwich. The enzyme substrate reaction produces a detectable signal proportional to the antigen concentration. In this approach, standard or test sample or control specimen and monoclonal antibodies specific for the analyte of interest conjugated to Horseradish Peroxidase (HRP-Conjugate) were introduced to polystyrene microwell strips pre-coated with monoclonal antibodies specific for the analyte of interest (p53 or bcl2 or SOD or 8OHdG). Chromogen solutions were added after incubation and washing, and the reaction stopped within a prespecified time. The ELx808 Absorbance Microplate Reader (BioTek Instruments Inc., Winooski, Vermont USA) was used to determine the optical density at 450 nm. The concentrations of analytes of interest in the samples were then extrapolated from standard curves using the optical density of each sample. Internal quality control was performed by analysing external reference samples together with the test samples. Human p53 ELISA kit (Catalogue No: EKHU-0309), Human bcl2 ELISA kit (Catalogue No: EKHU-0302), Human SOD ELISA kit (Catalogue No: EKHU-1058), and Human 8OHdG ELISA kit (Catalogue No: EKHU-1276) acquired from Melsin Medical Co., Limited, China were of the highest commercially analytical grade and had an intra-assay CV (%) < 10% and Inter-assay CV (%) < 15%.

## Measurement of serum MDA levels

The Malondialdehyde (MDA) colorimetric assay previously described by Buege & Aust [39] was used. In this method, the free MDA present in the sample which was a product of lipid peroxidation, reacted with Thiobarbituric Acid (TBA) to form an MDA-TBA adduct. The absorbance of the supernatant was measured on a spectrophotometer at $\lambda = 532$ nm. The MDA concentration was calculated using the regression equation of the MDA standard solution curve.

## Statistical analysis

Statistical Package for Social Scientists (SPSS Inc., USA) version 25.0 software was used for statistical analyses, including descriptive statistics. Continuous data were presented as mean ± standard error of mean (SEM), whereas categorical data were expressed as a frequency and percentage. The Kolmogorov Smirnov test was used to determine the normality of the variables. Parametric variables were compared using one-way analysis of variance (ANOVA), while chi-square or Fisher's exact test was used for categorical variables. The Pearson correlation test was used to examine the relationships between measured parameters. Multiple linear regression analysis was used to determine independent predictors of the regulators of apoptosis and cell cycle (p53 and bcl2 proteins) in the entire study population. All p values less than 0.05 were deemed statistically significant.

# Results

## Demographic characteristics

The demographic characteristics of study participants are presented in Fig 1 and Table 1. The participants' ages ranged from 18–65 years, with a mean ± SE of 38.35 ± 0.72 years, and they were found to have worked for an average of 10.57 ± 0.61 years. The mean duration of HIV

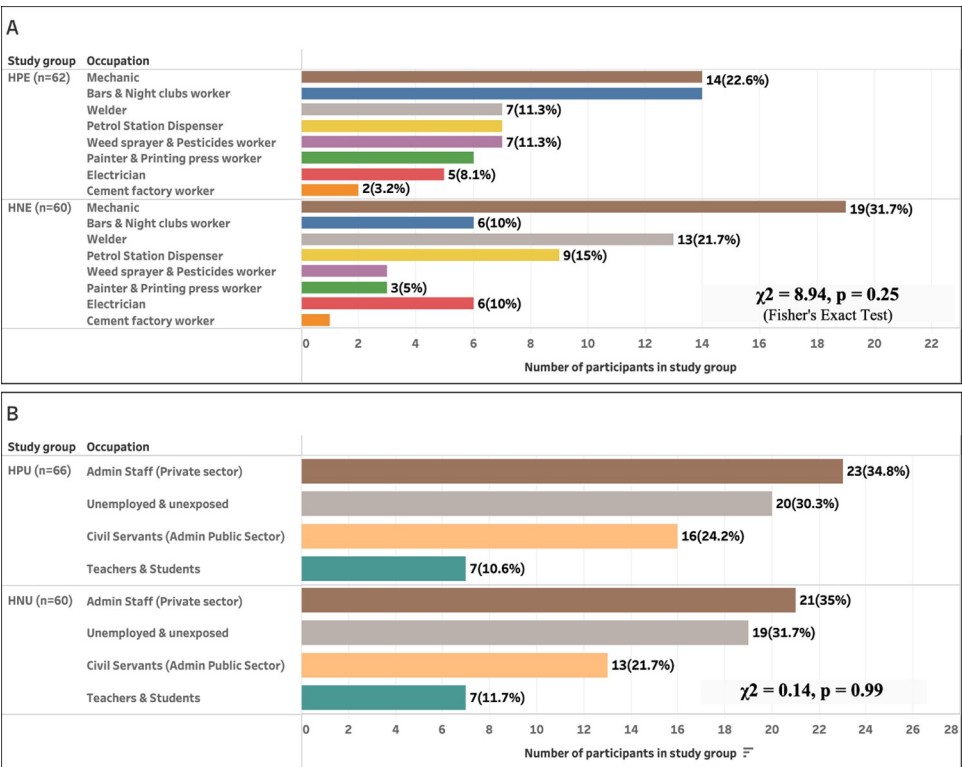

**Fig 1.** Comparable HIV status in study group: (A) Chemical-exposed job category and (B) Unexposed job category. A. *HPE (HIV positive exposed) vs HNE (HIV negative exposed). B. HPU (HIV positive unexposed) vs HNU (HIV negative unexposed).* χ2 = Chisquare.

**Table 1. Comparison of demographic characteristics of HIV-positive and HIV-negative participants within occupational categories.**

| Characteristics | Occupationally Exposed | | | | Occupationally Unexposed | | | |
|---|---|---|---|---|---|---|---|---|
| | HIV Positive (n = 62) | HIV Negative (n = 60) | χ2 | p-value | HIV Positive (n = 66) | HIV Negative (n = 60) | χ2 | p-value |
| Gender | | | 0.03 | 0.88 | | | 0.03 | 0.96 |
| Female, n(%) | 11(17.7%) | 10(16.7%) | | | 19(28.8%) | 17(28.3%) | | |
| Male, n(%) | 51(82.3%) | 50(83.3%) | | | 47(71.2%) | 43(71.7%) | | |
| Place of Residence | | | 0.41 | 0.52 | | | 4.22 | 0.06 |
| Rural, n(%) | 12(19.4%) | 9(15%) | | | 7(10.6%) | 9(15%) | | |
| Urban, n(%) | 50(80.6%) | 51(85%) | | | 59(89.4%) | 51(85%) | | |
| Current Employment Status | | | d | d | | | 0.03 | 0.87 |
| Employed, n(%) | 62(100%) | 60(100%) | | | 46(69.7%) | 41(68.3%) | | |
| Unemployed, n(%) | 0(0%) | 0(0%) | | | 20(30.3%) | 19(31.7%) | | |
| [a]Age (Years) | 38.84 ± 1.34 | 37.80 ± 1.44 | [b]0.53 | 0.60 | 39.36 ± 1.51 | 37.27 ± 1.43 | [b]1.00 | 0.32 |
| [a]Duration on occupation (years) | 9.94 ± 1.14 | 9.73 ± 1.06 | [b]0.13 | 0.90 | 12.70 ± 1.38 | 10.37 ± 1.34 | [b]1.21 | 0.23 |
| [a]BMI | 26.65 ± 0.56 | 26.12 ± 0.41 | [b]0.76 | 0.45 | 26.69 ± 0.30 | 25.62 ± 0.26 | [b]2.70 | 0.01* |

[a]Values are Mean ± SEM

[b]t statistics

* Statistically significant: p<0.05

χ2 = Chi-square: chi-square test done for categorical variables while t-test used for continuous variables.

d = No statistics were computed because 'Current Employment Status' for the occupationally exposed group is a constant.

exposure in the HIV positive group was 42.67 ± 2.45 months. The proportions of HIV-positive and HIV-negative participants were comparable when divided into jobs that predispose workers to chemical exposure (Fig 1A) and those that do not (Fig 1B). Age, gender, place of residence, current employment status, and duration of occupation were comparable between HIV positive and HIV negative individuals in the occupationally exposed and unexposed groups (Table 1). The occupationally exposed group in this study had worked across different occupations linked to toxic metals and hazardous chemicals. While BMI was comparable between HIV positive and HIV negative individuals in the occupationally exposed group, HIV positive individuals had a significantly higher BMI than HIV negative individuals in the occupationally unexposed group, $p < 0.01$.

## Levels of p53 and other biomarkers

The levels of apoptotic regulatory biomarkers wild-type p53 and bcl2, genotoxic biomarkers 8OHdG and MDA, as well as the antioxidant enzyme SOD and the immunological marker CD4, are presented in Fig 2. The molecular markers p53 and bcl2 decreased in the HPE group compared to the HPU, HNE, and HNU groups, respectively. While the HPE group had significantly lower p53 and bcl2 levels than the HNU group (p = 0.01 and p = 0.03 respectively), comparisons between the HPE group and the other two groups [HPU (p = 0.99 and p = 0.84 respectively) and HNE (p = 0.18 and p = 0.14 respectively)] were not significant. In contrast, the HPE group had higher 8OHdG levels than the HPU and HNE groups, although the difference was not statistically significant (p = 0.19 and p = 0.44 respectively). Additionally, SOD activity decreased in the HPE group relative to the HPU, HNE, and HNU groups, but the difference was not significant ($p > 0.05$). The MDA levels were significantly higher in the HPE group than in the HNE or HNU groups, respectively (p = 0.04 and p = 0.01 respectively). Additionally, the HPE group had higher MDA levels than the HPU group, but the difference was not significant ($p > 0.05$). Moreover, p53 and bcl2 levels were positively and significantly correlated with 8OHdG (r = 0.35, p<0.001; r = 0.36, p<0.001), SOD (r = 0.38, p<0.001; r = 0.39, p<0.001) whereas p53 and bcl2 levels were negatively but not significantly correlated with MDA (r = -0.03, p = 0.67; r = -0.05, p = 0.43) (Table 2).

## CD4 cell count—an indicator of immunological status

The immunological condition of the occupationally exposed and unexposed research participants is summarised in Fig 2 and Table 3 using the Centers for Disease Control and Prevention (CDC) CD4 count classification criteria of 500 cells/L; 200–499 cells/L; and 200 cells/L. In the occupationally exposed group, a significantly smaller proportion of HIV positive participants (54.8%) had CD4 cell counts ≥500cells/µL compared with 85% of HIV negative participants. Similarly, in the occupationally unexposed group, a significantly smaller proportion of HIV positive participants (50%) had CD4 cell counts ≥500cells/µL compared with 93.3% of HIV negative participants. Furthermore, the occupationally exposed HIV positive group (578.87 ± 33.64 cells/µL) and the HIV positive unexposed group (553.95 ± 36.86 cells/µL) had significantly lower CD4 counts than the HIV negative exposed (785.35 ± 36.8 cells/µL) and HIV negative unexposed (862.15 ± 43.29 cells/µL) groups (Fig 2). CD4 cell count was negatively and significantly correlated with MDA (r = -0.14, p = 0.03) (Table 2).

## Multiple linear regression for indicators of genotoxicity and apoptosis regulation

To examine the relationship between indicators of genotoxicity and apoptosis regulation using multiple linear regression, occupational chemical exposure, HIV status, CD4, SOD, 8OHdG,

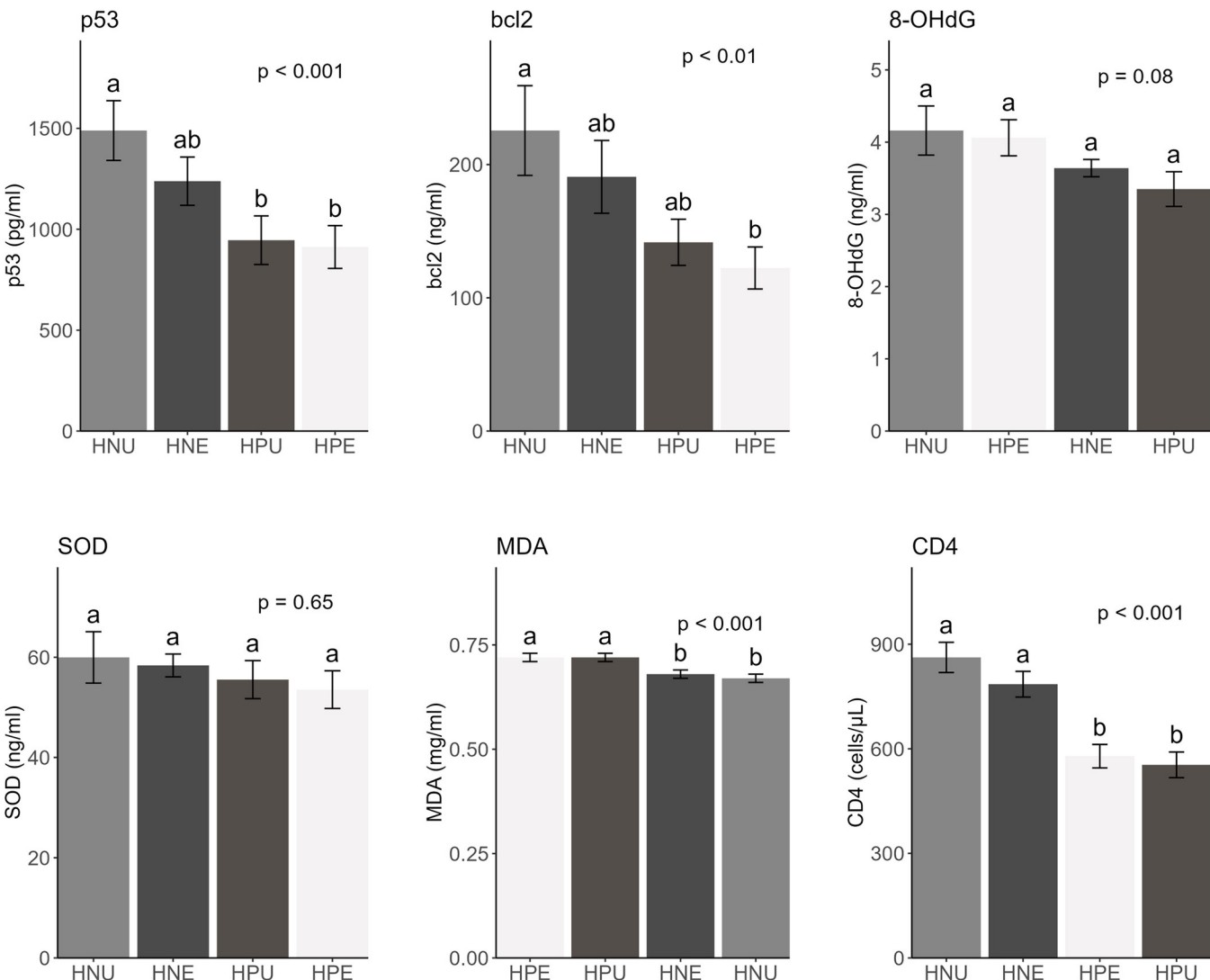

**Fig 2. Levels of p53 and other markers in HIV-negative unexposed (HNU), HIV-negative exposed (HNE), HIV-positive unexposed (HPU), and HIV-positive exposed (HPE) groups.** Bar graphs display the mean values of biomarkers p53, β-cell lymphoma (bcl2), 8-hydroxydeoxyguanosine (8-OHdG), superoxide dismutase (SOD), malondialdehyde (MDA), and CD4 counts, respectively, across study groups, with error bars indicating the standard error of the mean. In each plot, the bars are arranged in descending order of mean values for quick identification of trends in the data and clearer comparison between the different categories represented by the bars. Additionally, the overall p-value across the study group derived from a one-way ANOVA is indicated at the top of each figure. Pairwise comparison results (post hoc analysis) are represented by letters positioned above bars. When two bars do not share a common letter, it indicates a significant difference between study groups represented by the bars (p < 0.05).

and MDA were used as predictor variables, while apoptosis regulation markers, p53 and bcl2 were used as dependent variables. After adjusting for sex, age, BMI, and cigarette smoking, both HIV status and SOD activity were significantly associated with wildtype p53 and bcl2 (p < 0.05) (Table 4).

## Discussion

The co-occurrence of HIV infection and occupational exposure to hazardous chemicals poses a major concern in public health, as both factors may facilitate the concurrent degradation of both immunological and non-immune surveillance mechanisms against cancer. This study

**Table 2. Correlations between the evaluated biomarkers in the study population.**

|  | p53 | bcl2 | MDA | 8OHdG | SOD |
|---|---|---|---|---|---|
| p53 | 1 |  |  |  |  |
| bcl2 | 0.91** | 1 |  |  |  |
| MDA | -0.03 | -0.05 | 1 |  |  |
| 8OHdG | 0.35** | 0.36** | 0.01 | 1 |  |
| SOD | 0.38** | 0.39** | -0.07 | 0.76** | 1 |
| CD4 | 0.11 | 0.05 | -0.14* | 0.05 | 0.03 |

**. Correlation is significant at the 0.01 level (2-tailed).

*. Correlation is significant at the 0.05 level (2-tailed).

investigated the complex association between apoptotic regulatory markers and DNA oxidative response markers in the context of cancer susceptibility among individuals infected with HIV, who were also exposed to a cocktail of hazardous substances, including metals. The findings of this study indicate that p53 and bcl2 levels, which control apoptosis, were associated with HIV infection, oxidative damage, and a tendency for antioxidant depletion. Additionally, the decrease in wildtype p53 and bcl2 levels, as well as the increase in malondialdehyde levels, among HIV-positive individuals who had occupational exposure to chemical mixtures provides insight into the potential role of oxidative stress in viral and chemical carcinogenesis.

Individuals who were HIV-positive and had been exposed to chemical mixtures at work demonstrated notably reduced levels of p53 and bcl2 compared to HIV-negative persons who had not been exposed. This observation implies a plausible association between the co-occurrence of HIV infection and exposure to occupational chemicals, and a disruption in the apoptotic pathways. The proteins p53 and bcl2 are crucial in maintaining the equilibrium between cell survival and cell death [40]. Hence, the observed decrease in p53 and bcl2 levels could indicate an imbalance between cell proliferation and apoptosis in individuals infected by HIV and exposed to hazardous chemical agents, suggestive of a potential predisposition to malignancy. In a study that shares some similarities, Gruevska et al., [41] found that HIV patients had significantly lower levels of the tumour suppressor gene p53 than uninfected matched individuals and that this decreased expression of p53 was also observed at the protein level.

**Table 3. Immunological status of study groups based on CD4 cell counts.**

| [a]Immune Status | HIV Positive | HIV Negative | Test statistics | p-value |
|---|---|---|---|---|
| Occupationally Exposed |  |  | [b]13.35 | <0.001 |
| CD4 Count ≥500cells/μL | 34(54.8%) | 51(85%) |  |  |
| CD4 Count 200-499cells/μL | 26(41.9%) | 9(15%) |  |  |
| CD4 Count <200cells/μL | 2(3.2%) | 0(0%) |  |  |
| **Total n(%)** | **62(100%)** | **60(100%)** |  |  |
| Occupationally Unexposed |  |  | [b]30.67 | <0.001 |
| CD4 Count ≥500cells/μL | 33(50%) | 56(93.3%) |  |  |
| CD4 Count 200-499cells/μL | 24(36.4%) | 4(6.7%) |  |  |
| CD4 Count <200cells/μL | 9(13.6%) | 0(0%) |  |  |
| **Total n(%)** | **66(100%)** | **60(100%)** |  |  |

[a]Classification based on 2014 CDC Case Definition for HIV Infection Among Adolescents and Adults [37].

[b]Fisher's Exact Test Value

**Table 4. Effect of occupational exposure, HIV status and other biomarkers on p53 and bcl2.**

| | B | SE | Beta | t | sig. | Adjusted $R^2$ | [a]p-value |
|---|---|---|---|---|---|---|---|
| p53 | | | | | | 0.17 | <0.001* |
| Exposure | 94.30 | 118.91 | 0.05 | 0.79 | 0.43 | | |
| HIV status | 369.87 | 131.28 | 0.19 | 2.82 | 0.01* | | |
| CD4 | 0.09 | 0.20 | 0.03 | 0.48 | 0.63 | | |
| SOD | 9.48 | 3.05 | 0.29 | 3.11 | 0.00* | | |
| 8OHdG | 51.92 | 47.32 | 0.10 | 1.10 | 0.27 | | |
| MDA | 608.31 | 828.96 | 0.04 | 0.73 | 0.46 | | |
| bcl2 | | | | | | 0.18 | <0.001* |
| Exposure | 22.48 | 22.92 | 0.06 | 0.98 | 0.33 | | |
| HIV status | 69.00 | 25.31 | 0.18 | 2.73 | 0.01* | | |
| CD4 | -0.03 | 0.04 | -0.05 | -0.72 | 0.47 | | |
| SOD | 1.63 | 0.59 | 0.25 | 2.77 | 0.01* | | |
| 8OHdG | 15.61 | 9.12 | 0.16 | 1.71 | 0.09 | | |
| MDA | 15.55 | 159.79 | 0.01 | 0.10 | 0.92 | | |

Dependent Variables: p53 and bcl2

Predictors: Occupational chemical exposure, HIV status, CD4, SOD, 8OHdG, MDA

B = Unstandardized Coefficients; Beta = Standardized Coefficients

[a]Regression model p-value

* Statistically significant at $p < 0.05$

Malondialdehyde functions as a reliable marker for assessing lipid peroxidation and oxidative damage. In this study, occupationally exposed HIV-positive individuals had increased MDA levels, indicating that individuals infected with HIV and exposed to occupational chemicals experience elevated levels of oxidative stress. This finding is consistent with the literature suggesting that exposure to certain chemicals in the workplace, such as toxic metals, may cause oxidative stress, leading to further damage to cellular constituents, including DNA [42, 43]. Indeed, the presence of HIV infection may create an enabling environment favourable for cancer development and progression, precipitated by immune suppression as well as processes such as DNA mutations and defective DNA repair, which can be attributed to mitochondrial dysfunction and elevated oxidative stress [44].

Oxidative stress in occupationally exposed HIV-infected individuals, may contribute to the impairment of p53 function, potentially leading to the initiation or progression of carcinogenesis. Although p53 is generally activated in response to DNA damage [45], chronic exposure to occupational carcinogens may result in prolonged activation and ultimately depletion of p53. Therefore, the decreased wildtype p53 in response to chronic occupational chemical exposure and HIV infection observed in this study may compromise the critical role of p53 in preserving DNA integrity and preventing cancer. This agrees with studies demonstrating that people living with HIV or exposed to chemical carcinogens showed increased genomic instability [16, 46]. Oxidative stress plays a critical role in chemical and viral carcinogenesis [47, 48].

Multiple factors could potentially influence the susceptibility of cells to cancer. The observed association between HIV status, 8OHdG and SOD, with levels of wildtype p53 and bcl2 suggest that these indicators are influenced not only by individual exposures, but also by the combined factors altering cellular responses to oxidative stress and apoptosis. The complex relationship between these factors highlights the diverse and intricate mechanisms that contribute to the risk of cancer in people living with HIV, but also exposed to chemical toxins. This observation is consistent with previous studies that proposed a relationship between HIV

infection and oxidative stress, which can result in changes to cellular mechanisms, such as apoptotic pathways [44, 49]. Darbinian *et al.*, [50] demonstrated that initiation of apoptotic responses was followed by elevated mtDNA damage in cells treated with HIV-1 proteins or infected with HIV-1. Conversely, the accumulation of damaged DNA can result from abnormal or failure of apoptosis which may lead to cancer [51].

A decreased CD4 cell count was observed in HIV-infected individuals regardless of occupational chemical exposure. This aligns with the prevailing consensus that HIV infection impairs the immune system by preferentially destroying CD4-positive T cells. Indeed, the altered immunological profiles of the HIV-positive study population are consistent with previous research on the detrimental effect of HIV infection on immune system functionality [52, 53]. Moreover, the observation of a negative correlation between CD4 count and MDA concentrations provides further evidence for the probable influence of both HIV infection and chemical exposures on cellular pathways associated with oxidative stress and apoptosis. Our findings suggest that impaired immune surveillance indicated by low CD4 cell count may impede the body's capacity to identify and eradicate malignant or pre-malignant cells and therefore play a role in increased cancer risk among HIV-infected workers. Furthermore, these findings are consistent with evidence indicating that oxidative stress plays a pivotal role in the interconnection between viral infections, hazardous exposures, and the development of cancer [54–56]. This may potentially interact with chemical exposures, leading to an increased vulnerability to cancer. Kang *et al.* [57] demonstrated that an intact CD4(+) T-cell-mediated adaptive immune response is important for tumour immune surveillance.

The present study highlights the complex relationships among apoptotic regulatory indicators, DNA oxidative response markers, and the risk of cancer in a distinct population of HIV-infected individuals who have been occupationally exposed to hazardous chemicals. Notably, the current study underscored the influence of both HIV infection and chemical exposures on cellular pathways associated with oxidative stress and apoptosis. The majority of previous studies focused on the influence of HIV infection or occupational exposure on oxidative stress indicators in isolation [56, 58]. However, it is imperative to acknowledge the study's limitations. The cross-sectional design makes it difficult to establish causal relationships between occupational chemical exposure, HIV infection, and changes in levels of cancer risk biomarkers studied. Secondly, the study did not investigate specific cancer types or long-term cancer outcomes, which could have provided valuable insight into the clinical implications of these findings. Longitudinal studies are advocated to provide more robust evidence.

## Conclusion

Immune deficiency, antioxidant deficits, oxidative stress, and dysregulation of apoptosis contribute to increased genomic instability and elevated cancer risk in HIV-infected individuals occupationally exposed to chemical mixtures. In addition, our findings suggest that optimal levels of the enzymatic antioxidant SOD may be essential for p53 to play its protective role in maintaining genomic stability. These findings highlight the need to design and implement preventive interventions to reduce the risk of viral and chemical carcinogenesis among the vulnerable population of HIV-infected individuals exposed to occupational chemical hazards.

## Acknowledgments

The authors extend their gratitude to the study's community outreach team, led by Mr. Saviour Nkanu, for their support in participant recruitment and to all participants involved in this study.

## Author Contributions

**Conceptualization:** Donald C. Udah, Adeleye S. Bakarey, Olusegun G. Ademowo, John I. Anetor.

**Data curation:** Donald C. Udah, Adeleye S. Bakarey, Maxwell Omabe, Victory F. Edem.

**Formal analysis:** Donald C. Udah, Gloria O. Anetor, Olusegun G. Ademowo, John I. Anetor.

**Funding acquisition:** Donald C. Udah, John I. Anetor.

**Investigation:** Donald C. Udah, Adeleye S. Bakarey, Gloria O. Anetor, Victory F. Edem, Olusegun G. Ademowo, John I. Anetor.

**Methodology:** Donald C. Udah, Adeleye S. Bakarey, Gloria O. Anetor, Maxwell Omabe, Victory F. Edem, Olusegun G. Ademowo, John I. Anetor.

**Supervision:** Olusegun G. Ademowo, John I. Anetor.

**Visualization:** Donald C. Udah, Maxwell Omabe.

**Writing – original draft:** Donald C. Udah.

**Writing – review & editing:** Donald C. Udah, Adeleye S. Bakarey, Gloria O. Anetor, Maxwell Omabe, Victory F. Edem, Olusegun G. Ademowo, John I. Anetor.

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
