## [Decision Letter · Decision Letter 0]

29 Mar 2024

PGPH-D-23-02537

Occupational cancer risk surveillance in HIV-infected individuals exposed to chemicals: Role of p53 molecular marker

Dear Dr. Udah,

Thank you for submitting your manuscript to PLOS Global Public Health. After careful consideration, we feel that it has merit but does not fully meet PLOS Global Public Health’s publication criteria as it currently stands. Therefore, we invite you to submit a revised version of the manuscript that addresses the points raised during the review process.

The reviewer has outlined some major concerns, especially around the methodological details and results presented.

Please note that we have only been able to secure a single reviewer to assess your manuscript. We are issuing a decision on your manuscript at this point to prevent further delays in the evaluation of your manuscript. Please be aware that the editor who handles your revised manuscript might find it necessary to invite additional reviewers to assess this work once the revised manuscript is submitted. However, we will aim to proceed on the basis of this single review if possible. 

We look forward to receiving your revised manuscript.

Kind regards,

Annesha Sil, Ph.D.

PLOS Staff Editor

Journal Requirements:

1. Please ensure that Funding Information and Financial Disclosure Statement are matched.

2. In the Funding Information you indicated that no funding was received. Please revise the Funding Information field to reflect funding received.

Additional Editor Comments (if provided):

Reviewers' comments:

Reviewer's Responses to Questions

**Comments to the Author**

1. Does this manuscript meet PLOS Global Public Health’s publication criteria? Is the manuscript technically sound, and do the data support the conclusions? The manuscript must describe methodologically and ethically rigorous research with conclusions that are appropriately drawn based on the data presented.

Reviewer #1: Yes

2. Has the statistical analysis been performed appropriately and rigorously?

Reviewer #1: Yes

3. Have the authors made all data underlying the findings in their manuscript fully available (please refer to the Data Availability Statement at the start of the manuscript PDF file)?

Reviewer #1: Yes

4. Is the manuscript presented in an intelligible fashion and written in standard English?

Reviewer #1: Yes

5. Review Comments to the Author

Reviewer #1: General Comment: The title of the manuscript did not reflect the entirety of the work done. There should be a way to ensure that the entirety of the work done is captured to give correct first-sight impression about the study.

Specific comments:

Line 34: The use of the word “Apparently healthy adults” to include HIV patients may be inappropriate. The statement should read “Study Participants or Subjects…).

Line 35: bcl2 should be written in full, followed by being put in bracket for subsequent use

Lines 65-66: “..such as Kaposi sarcoma and non-Hodking’s lymphoma” can be moved to line 61, after “ AIDS -defining cancers…” to actually define the AIDS-defining cancers are.

Line 101: the interplay of that of bcl2 was not well elucidated like that of p53.

Line 107: It will be great if the description of “the community” and the “the health facility” can be stated, to define the study site.

Line 111: ‘and’ all recruited into the study

Line 113: replace “elicit” with “obtain”

Lines 120 – 123: It will be good if the nature of toxicants that the subjects were exposed to, based on their occupations, are mentioned here, then supported by the references.

Lines 127 - 132: Please reframe this paragraph. Perhaps you start with “participants with history of hereditary …… were excluded from this study”.

Line 138: Please state the quality control measures taken, to achieve accurate anthropometric measurements.

Line 147: Please be specific with the centrifugation process. The centrifuge speed cannot be ranged.

Line 153-154: please reframe the statement. It is not explicit enough

Table 1 refers:

It will be better if another column is created to express the tools of the numerical variables (the t statistic). It is not actually clear with the way it is presently expressed.

Lines 203 -205: Seem tautologic. Remove and report the BMI outcome between HIV positive and Negative

Lines 206 – 208: Should preceded the sociodemographic descriptions in line 197.

Figure 2 refers: The figures did not adequately display the proper interpretation of the outcomes. Only the F-value could be observed. The bars showing individual comparisons within and between groups (post-hoc) are not displayed. This should be corrected. All labelled bars should be properly annotated.

Line 256: The said bar chart in Figure 2 could not be found. I am of the opinion that subsequent tables and figures are displayed separately from Figure 2. Missing figures could be clearly seen because of the much figures expressed already, with no distinct referral labels to them.

Table 3 : The title “Immunological Status” is a broad statement. It could be put as “Immunological Status of Study Groups based on the CD4 Levels”.

Line 299: “compared to” instead of “in comparison to”

Line 322: The first statement required a reference.

Line 362: Replace the word “prior” with “previous”

6. PLOS authors have the option to publish the peer review history of their article (what does this mean?). If published, this will include your full peer review and any attached files.

**Do you want your identity to be public for this peer review?** For information about this choice, including consent withdrawal, please see our Privacy Policy.

Reviewer #1: No

---

## [Decision Letter · Decision Letter 1]

29 May 2024

PGPH-D-23-02537R1

Increased cancer risk in HIV-infected individuals occupationally exposed to chemicals: depression of p53 as the key driver

Dear Dr. Udah,

Thank you for submitting your manuscript to PLOS Global Public Health. After careful consideration, we feel that it has merit but does not fully meet PLOS Global Public Health’s publication criteria as it currently stands. Therefore, we invite you to submit a revised version of the manuscript that addresses the points raised during the review process.

We look forward to receiving your revised manuscript.

Kind regards,

Kazeem Sanjo Akinwande, Ph.D

Guest Editor

Journal Requirements:

Additional Editor Comments (if provided):

Reviewers' comments:

Reviewer's Responses to Questions

**Comments to the Author**

1. If the authors have adequately addressed your comments raised in a previous round of review and you feel that this manuscript is now acceptable for publication, you may indicate that here to bypass the “Comments to the Author” section, enter your conflict of interest statement in the “Confidential to Editor” section, and submit your "Accept" recommendation.

Reviewer #2: (No Response)

2. Does this manuscript meet PLOS Global Public Health’s publication criteria? Is the manuscript technically sound, and do the data support the conclusions? The manuscript must describe methodologically and ethically rigorous research with conclusions that are appropriately drawn based on the data presented.

Reviewer #2: Yes

3. Has the statistical analysis been performed appropriately and rigorously?

Reviewer #2: Yes

4. Have the authors made all data underlying the findings in their manuscript fully available (please refer to the Data Availability Statement at the start of the manuscript PDF file)?

Reviewer #2: No

5. Is the manuscript presented in an intelligible fashion and written in standard English?

Reviewer #2: Yes

6. Review Comments to the Author

Reviewer #2: 1. Figures 1A and 1B were not sighted in the MSW document which was reviewed

2. Under Findings, p53 metabolism, and conclusions drawn from this study are consistent with previously published work on HIV pathogenesis in relation to p53. What is new? This should be highlighted in the manuscript

3. Authors should clarify if the study is strictly a cross-sectional study. Described methodology has component of case-control design. Check and reconcile this. If it has a mixed- method design explain same

4. Was the questionnaire validated prior to use for the study? Mention same if it was . If it was not, validation will be required

5. How did authors ascertain validity of electronic measuring scale? Was the reading taken by one or multiple persons? How did you ascertain inter and intra variation validity .

6. Principle of all chemical based measurements should be explained first before description of procedure

7. Write in full first all abbreviations mentioned in the manuscript .

7. PLOS authors have the option to publish the peer review history of their article (what does this mean?). If published, this will include your full peer review and any attached files.

**Do you want your identity to be public for this peer review?** For information about this choice, including consent withdrawal, please see our Privacy Policy.

Reviewer #2: No

---

## [Editor Report · Decision Letter 2]

24 Jun 2024

Increased cancer risk in HIV-infected individuals occupationally exposed to chemicals: depression of p53 as the key driver

PGPH-D-23-02537R2

Dear Mr. Udah,

We are pleased to inform you that your manuscript 'Increased cancer risk in HIV-infected individuals occupationally exposed to chemicals: depression of p53 as the key driver' has been provisionally accepted for publication in PLOS Global Public Health.

Best regards,

Kazeem Sanjo Akinwande, Ph.D

Guest Editor